

# Environmental awareness and sustainable consumption behaviours of Turkish nursing students

Renginar Öztürk Dönmez[1] and Eda Yardımcı[2]

[1] Faculty of Nursing, Ege University, İzmir, Türkiye
[2] Health Science Institute, Ege University, İzmir, Türkiye

## ABSTRACT

**Objectives:** This study aims to determine the environmental awareness and sustainable consumption behaviours of nursing students.

**Methods:** This cross-sectional study was conducted with 380 undergraduate nursing students studying at Faculty of Nursing, Ege University, Türkiye. The Personal Information Form, Environmental Awareness Scale (EAS), and Sustainable Consumption Behaviours Scale (SCBS) were used to collect the research data. Number, percentage, mean, Kolmogorov-Smirnov Z test, t-test, one-way analysis of variance, Pearson's correlation test, and linear regression analysis were used in statistical evaluation.

**Results:** The mean total score on the EAS was 46.12 (SD = 6.60), and the mean total score of the SCBS was 61.83 (SD = 9.10). A significant, moderate, and positive correlation was found between students' environmental awareness and sustainable consumption behaviours. Moreover, students' gender and the people they lived with caused statistically significant differences in the EAS score. The people students lived with, the place where they lived for the longest time, and their perceived income level caused statistically significant differences in the SCBS score.

**Conclusions:** Nursing students had high environmental awareness and moderate to good sustainable consumption behaviours. It is necessary to include the contribution of sustainable practices to the environment and the effects of these practices on human health in curricula for nurses and thus provide environmental awareness to students.

Corresponding author
Renginar Öztürk Dönmez,
renginar.ozturk.donmez@ege.edu.tr

## INTRODUCTION

The world population is expected to reach nine billion by 2050. Rapid population growth increases the production and consumption cycle, and the use of health services and causes the deterioration of planetary health (*Lenzen et al., 2020*). In 2015, the 193 countries that make up the United Nations (UN) agreed to adopt the 2030 Agenda for Sustainable Development to eliminate poverty, protect planet Earth, combat climate change and ensure that all people live in prosperity. The 12th goal, one of Sustainable Development Goals (SDGs), is responsible production and responsible consumption. Responsible production and responsible consumption are an environmentally responsible behaviour

that defends nature and protects ecology (*Yue et al., 2020*). Responsible production and responsible consumption include the elements of purchasing recyclable and green/environmental products using resources effectively, using energy economically, and respecting nature during the production phase. Using cleaners that are less harmful to the environment, not buying unnecessary products, choosing electronic devices that consume less electrical energy, and reuse papers include responsible production and responsible consumption behaviours (*Doğan, Bulut & Kökalan Çımrın, 2015*). In the goal of responsible production and consumption, it is emphasized that we urgently need to reduce our ecological footprint by achieving economic growth and sustainable development and by changing the way we produce and consume goods and resources (*Katila et al., 2019*). Türkiye ranks 18th in the world in terms of population size, with approximately 85 million people (*World Bank, 2024*). In Türkiye, rapid population growth leads to excessive consumption of natural resources and aggravated deterioration of the ecological environment. The SDGs are also being developed in Türkiye herewith. The UN's "2nd National Review of the SDGs" for Türkiye specified a shared vision and laid the foundation for achieving common goals (*Kurt & Özbaş, 2023*).

Healthcare institutions deliver services 24 h and 7 days. Therefore, these are areas with immense energy consumption, waste production and intense environmental impacts (*Sapuan et al., 2022*). Healthcare systems are responsible for 4–5% of the emissions of greenhouse gases that cause climate change worldwide (*Rodríguez-Jiménez et al., 2023*). The delivery of healthcare influences on the environment and contributes to climate change. While the SDGs reveal the relationship between individual and planetary health, they also provide guidance to nurses regarding their interventions (*Dossey, Rosa & Beck, 2019*). Nurses, who constitute the majority of healthcare systems, are key stakeholders in advancing health across the globe. Nurses can engage and advance the UN 2030 the SDGs Agenda and take charge of change. Through strategic approaches to obtaining the SDGs targets, nurses will be able to improve quality of life for themselves and the public they serve (*Rosa et al., 2019*; *Li et al., 2021*). The annual statement of the *International Council of Nursing (ICN) (2018)* and the *American Nurses Association (ANA) (2008)* emphasized the need for nurses to take immediate action in order to establish climate-resilient health systems. The ICN has determined the 2017 theme as "Nurse: Leading Voice–Achieving SDGs" (*Kıvanç et al., 2020*). Nurses and nursing students are expected to act as key advocates of sustainable production and consumption (*Kurt & Özbaş, 2023*; *Sherman et al., 2023*). Despite these developments, it is still not specified the number of credits or hours of teaching on sustainability and health across Europe and Türkiye. Some universities in a few countries, the United Kingdom, Germany, Spain, Netherlands, Belgium and Sweden, have embedded as a cross-curricular theme some subjects (*Huss et al., 2020*; *Shaw et al., 2021*). In the Turkiye nursing core curriculum, the number of credits or hours is not defined for the relationship between sustainability, environment and health. Only the outcomes of being environmentally literate and knowing the effects of the environment on health are included (*Türkiye Nursing Core Education Curriculum, 2022*). *Şimşek & Erkin (2022)* stated that 63.8% of nurse students have not heard of the concept of sustainability and 77.4% do not know about the SDGs. If SDGs are to be achieved, nurses

must have knowledge and act accordingly about SDGs targets. Therefore, it is necessary to realize the connection between the responsible production and consumption, sustainability and health and the responsibility to change the *status quo* (*Lopez-Medina et al., 2019*). In order to environmental and sustainability awareness to be reflected in clinical practices in the coming years, there is a need to determine the environmental awareness and sustainable consumption behaviours of nursing students (*Cruz, Alshammari & Felicilda-Reynaldo, 2018*). It has been emphasized in some studies that nursing students are inadequately prepared to understand the relationships between the use of resources, sustainability and health (*Lopez-Medina et al., 2019*; *Álvarez-Nieto et al., 2022*). Although there are a very limited number of studies examining the sustainable consumption attitudes and behaviours of nursing students in different countries, there are no studies conducted with nursing students in Türkiye (*Cruz, Alshammari & Felicilda-Reynaldo, 2018*; *Lopez-Medina et al., 2019*; *Álvarez-Nieto et al., 2022*). Incorporating requirements into the curriculum in line with the results obtained from research offers a useful and new approach both for the benefit of the patient and the development of the skills of professional members (*Huss et al., 2020*). However, there is a gap in the literature about the environmental awareness and sustainable consumption behaviours of nursing students that will provide evidence for this. This study aims to determine the environmental awareness and sustainable consumption behaviours of Turkish nursing students.

The research hypotheses were as follows:

$H1_1$: Nursing students have high environmental awareness.

$H1_2$: Nursing students have high sustainable consumption behaviours.

$H1_3$: There is a relationship between the environmental awareness level and the sustainable consumption behaviours levels of nursing students.

$H1_4$: Sociodemographic characteristics of nursing students affect their level of environmental awareness.

$H1_5$: Sociodemographic characteristics of nursing students affect their level of sustainable consumption.

## MATERIALS AND METHODS

### Study design and sampling

This cross-sectional study was conducted at Ege University Faculty of Nursing located in the west of Türkiye between July and September 2021. Undergraduate nursing students of nursing faculty constituted the study population ($N = 1,193$). To determine the sample in the study, the sample formula with the known population was used ($n = Nt^2pq/d^2(N - 1) + t^2pq$) (*Erdoğan, Nahcivan & Esin, 2014*).

The reference regarding the incidence of sustainable consumption behaviour (42%) was taken from the study results obtained by *Kukkonen, Kärkkäinen & Keinonen (2018)*. ($n = 1,193 \times 1.96^2 \times 0.42 \times 0.58/0.05^2 \times 1,192 + 1.96^2 \times 0.42 \times 0.58$). With the calculation made accordingly 95% confidence interval, the sample size was determined to be at least 225 participants. The current study was completed with 380 volunteer undergraduate nursing students using a purposive sampling method. The Strengthening the Reporting of

Observational Studies in Epidemiology (*STROBE, 2023*) checklist was used in the study design and drafting of the manuscript.

## Instruments and data collection

The data were collected through online forms using the Personal Information Form, Environmental Awareness Scale (EAS), and Sustainable Consumption Behaviours Scale (SCBS) (https://forms.gle/KSU3ete76zPHw9Ja6) after obtaining ethical approval from the Ethics Committee of Ege University (Date: June 25, 2021, Number: E.208620). The authors have permission to use the instruments from the copyright holders. The voluntary participation of nursing students was questioned before implementing the form. Nursing students who approved it answered the questions. The online instruments helped ensure that there were no missing data in the submitted responses. It took approximately 30 min to complete the instruments.

The personal information form consists of 12 questions to identify the socio-demographic characteristics of nursing students (*Richardson et al., 2016*; *Ntanos et al., 2018*; *Lee et al., 2019*; *Shaw et al., 2021*).

The EAS developed by *Alkaya et al. (2016)* consists of 11 questions. Three subscales were obtained as a result of the factor analysis. The subscales are ecological awareness, self-awareness, and behavioural awareness. It is a five-point Likert scale and is rated as 1: Strongly disagree, 2: Disagree, 3: Neither agree nor disagree, 4: Agree, 5: Strongly agree. The questions consist of statements to measure participants' environmental awareness and green product purchasing status. Cronbach's alpha value of the scale was found to be 0.815. Individual environmental awareness increases with the increasing score (*Alkaya et al., 2016*). In our study, Cronbach's alpha value of the scale was found to be 0.901. The Kolmogorov–Smirnov test was performed to examine the normal distribution suitability of the scale's mean score. It was revealed that the scale score was suitable for normal distribution ($p > 0.05$).

The SCBS was developed by *Doğan, Bulut & Kökalan Çımrın (2015)* and consists of 17 questions. The SCBS consists of four subscales. The subscales were defined as Environmental Awareness (Five items), Out-of Need Purchasing (Five items), Saving (Four items), and Reusability (Three items). Items in the environmental awareness sub-dimension include purchasing cleaning agents that are less harmful to the environment, purchasing clothes made of natural materials, paying attention to the fact that purchased can be degraded in nature, warning friends to raise awareness, and supporting companies that sell environmentally friendly products. Items in the Out-of-Need Purchasing sub-dimension include buying electronic devices, buying clothes, buying products off the list while grocery shopping, and stocking up on food, even if one does not perceive the need. Savings sub-dimension items include purchasing energy-saving electronic devices, using energy-saving light bulbs, and taking into account the amount of electricity consumption when purchasing. The reusability sub-dimension items include renting instead of purchasing products, recycling products such as cardboard and tin, and reusing used papers. The measurement tool with a five-point Likert rating is scored as Never (1), Very Rarely (2), Sometimes (3), Usually (4), and Always (5). Reverse scoring is

made in the subscale of Out-Of Need Purchasing. In the SCBS, sustainable consumption behaviours increase to the same extent as the score increases. Its Cronbach's alpha coefficient is 0.771 (*Doğan, Bulut & Kökalan Çımrın, 2015*). In our study, Cronbach's alpha value of the scale was found to be 0.835. The Kolmogorov-Smirnov test was conducted to examine the normal distribution suitability of the scale's mean score. It was found that the scale score was suitable for normal distribution ($p > 0.05$).

### Measurement

The dependent variables of the study are the EAS and the SCBS total scores. The independent variables of the study were nursing students' age, gender, class level, type of high school they graduated from, the people they lived with, the place where they lived for the longest time, their income level, and membership in any environmental organization.

### Data analysis

Number, percentage, mean and standard deviation were used in the analysis of sociodemographic variables. Means and standard deviations were used in evaluation of scales, and whether the dependent variable showed a homogeneous distribution was analyzed with the Kolmogorov-Smirnov Z test. One-way analysis of variance, independent t-test, Pearson's correlation test, Dunnett's t-test, and linear regression, which are parametric tests, analysis were used as hypothesis tests. The statistical significance level was considered as $p < 0.05$ at a confidence interval of 95%.

## RESULTS

The mean age of nursing students was 21.88 (SD = 2.72). Other sociodemographic characteristics of nursing students are shown in Table 1.

The mean EAS was 46.12 (SD = 6.60). It was determined that nursing students scored 4.47 (SD = 0.80) points on average on the ecological awareness subscale, 4.18 (SD = 0.79) points on average on the self-awareness subscale, and 3.85 (SD = 0.82) points on average on the behavioural awareness subscale.

Gender of nursing students ($p = 0.024$) and the people they lived with ($p = 0.021$) caused a statistically significant difference in the mean score on the EAS. Higher EAS scores were found in female nursing students compared to male, nursing students living with their families compared to those living alone or with friends. Nursing students' age, class level, type of high school they graduated from, the place where they lived for the longest time, and perceived income level were the independent variables not cause a difference (Table 2).

The linear regression model created to identify the independent variables predicting the environmental awareness levels of nursing students found no autocorrelation between the variables. The created model was found to be significant, linear, and good (F = 33.759, $p = 0.001$, Durbin Watson = 2.227). It was revealed that 21.3% of the change in the environmental awareness levels of nursing students was related to changes in gender ($p = 0.016$), the people they lived with ($p = 0.041$), and their sustainable consumption behaviours level ($p = 0.001$). It was found that the standard deviation change in

**Table 1** The socio-demographic characteristics of the nursing students (*n* = 380).

| Characteristics | *n* | % |
| --- | --- | --- |
| **Age group** | | |
| 17–22 | 207 | 54.5 |
| 22–26 | 173 | 45.5 |
| **Gender** | | |
| Female | 269 | 70.8 |
| Male | 111 | 29.2 |
| **Class** | | |
| 1 | 101 | 26.6 |
| 2 | 107 | 28.2 |
| 3 | 85 | 22.4 |
| 4 | 87 | 22.9 |
| **Graduated high school** | | |
| General high school | 296 | 77.9 |
| Science and technical high school | 84 | 22.1 |
| **The people they lived with** | | |
| Parent | 339 | 89.2 |
| Friend/alone | 41 | 10.8 |
| **The place where they lived for the longest period of time** | | |
| Village | 59 | 15.5 |
| Town | 125 | 32.9 |
| City | 47 | 12.4 |
| Metropolitan city | 149 | 39.2 |
| **Perceived income level** | | |
| Income less than expenses | 81 | 21.3 |
| Income equal to expenses | 260 | 68.4 |
| Income more than expenses | 39 | 10.3 |

**Table 2** Comparison of the mean total scores of the environmental awareness scale and the sustainable consumption behaviors scale based on nursing students' characteristics (*n* = 380).

| Characteristics | Environmental awareness scale | | | Sustainable consumption behaviors scale | | |
| --- | --- | --- | --- | --- | --- | --- |
| | M (SD) | t/F* | *p* | M (SD) | t/F* | *p* |
| **Age group** | | | | | | |
| 17–22 | 46.24 (5.97) | 0.314 | 0.754 | 60.85 (9.58) | 1.931 | |
| 22–26 | 46.02 (7.09) | | | 62.66 (9.61) | | 0.056 |
| **Gender** | | | | | | |
| Female | 46.71 (5.54) | 2.282 | 0.024 | 61.93 (8.98) | 0.331 | 0.740 |
| Male | 44.71 (8.51) | | | 61.59 (9.42) | | |
| **Grade** | | | | | | |
| 1 | 46.51 (7.32) | | | 61.45 (10.52) | 1.148 | |
| 2 | 45.46 (6.84) | 0.514 | 0.673 | 60.82 (8.69) | | 0.329 |

| Characteristics | Environmental awareness scale | | | Sustainable consumption behaviors scale | | |
|---|---|---|---|---|---|---|
| | M (SD) | t/F* | p | M (SD) | t/F* | p |
| 3 | 46.28 (6.26) | | | 62.25 (8.25) | | |
| 4 | 46.33 (5.72) | | | 63.13 (8.54) | | |
| **Graduated high school (HS)** | | | | | | |
| General HS | 46.29 (6.71) | 0.910 | 0.364 | 62.31 (8.94) | 1.913 | 0.471 |
| Science and technical HS | 45.54 (6.21) | | | 60.16 (9.49) | | |
| **The people they lived with** | | | | | | |
| Parent | 46.53 (5.98) | 2.400 | 0.021 | 62.29 (8.78) | 2.830 | |
| Friend/alone | 42.75 (9.86) | | | 58.07 (10.74) | | 0.005 |
| **The place where they lived for the longest time** | | | | | | |
| Village | 45.96 (7.83) | | | 62.94 (10.06) | | 0.001 |
| Town | 46.21 (5.31) | 0.500 | 0.682 | 60.61 (9.20) | 5.777 | d > a, a > b, |
| City | 45.10 (7.02) | | | 58.06 (9.68) | | b > c |
| Metropolitan city[d] | 46.43 (6.93) | | | 63.59 (7.93) | | (0.001)** |
| **Perceived income level** | | | | | | |
| Income < expenses[a] | 46.12 (7.86) | | | 62.39 (8.24) | 3.671 | 0.026 |
| Income = expenses[b] | 46.31 (6.21) | 0.777 | 0.461 | 62.22 (7.68) | | c < a,b |
| Income > expenses[c] | 44.89 (6.28) | | | 58.12 (7.85) | | (0.015)** |

Notes:
M, Mean; SD, Standard deviation.
* t, Independent t test; F, One way ANOVA.
** Dunnett t test.

**Table 3 The predictive variables on environmental awareness of the nursing students ($n = 380$).**

| Dependent variable | Independent variable | B | SE | t | p | Tolerance | VIF |
|---|---|---|---|---|---|---|---|
| **Environmental awareness** | Constant | 31.761 | 2.697 | 12.268 | 0.001 | | |
| | Gender | 1.636 | 0.679 | −2.260 | 0.016 | 0.956 | 1.036 |
| | The people they lived with | 2.052 | 1.001 | 2.026 | 0.041 | 0.945 | 1.058 |
| | SCBS total score | 0.303 | 0.034 | 8.739 | 0.001 | 0.979 | 1.021 |
| | Model | R = 0.462 Adj. $R^2$ = 0.213 Durbin Watson = 2.227 | | | | | |
| | | F = 33.759 $p = 0.001$* | | | | | |

Note:
* Linear regression analysis.
SE, Standard error; VIF, Variance Inflation Factor; F, One way ANOVA.

environmental awareness level was due to the 1.636 change in gender, the 2.052 change in the people they lived with and the 0.303 change in the SCBS total score (Table 3).

The mean SCBS was 61.83 (SD = 9.10). The mean score on the environmental awareness subscale was 3.24 (SD = 0.97), the mean score on the out-of-need purchasing subscale was 2.20 (SD = 0.84), the mean score on the saving subscale was 3.97 (SD = 0.97), and the mean score on the reusability subscale was 3.54 (SD = 1.04).

The people nursing students lived with ($p = 0.005$), the place where they lived for the time ($p = 0.001$), and their perceived income level ($p = 0.026$) caused a statistically

**Table 4 The predictive variables on sustainable consumption behaviors of the nursing students (*n* = 380).**

| Dependent variable | Independent variable | B | SE | t | *p* | Tolerance | VIF |
|---|---|---|---|---|---|---|---|
| **Sustainable consumption behaviors** | | | | | | | |
| | Constant | 38.576 | 3.883 | 9.931 | 0.001 | | |
| | The people they lived with | 1.953 | 1.384 | 1.411 | 0.159 | 0.951 | 1.051 |
| | The place where they lived for the longest | 0.649 | 0.375 | 1.731 | 0.084 | 0.973 | 1.021 |
| | Perceived income level | 1.455 | 0.771 | 1.887 | 0.060 | 0.972 | 1.029 |
| | EAS total score | 0.572 | 0.065 | 8.864 | 0.001 | 0.967 | 1.034 |
| | Model | R = 0.452 Adj. $R^2$= 0.204 Durbin Watson = 1.946 F = 23.967 *p* = 0.001* | | | | | |

Note:
* Linear regression analysis.
SE, Standard error; VIF, Variance Inflation Factor; F, One way ANOVA.

significant difference in the mean score on the SCBS. Nursing students living with their families compared to those living alone or with friends, those living in metropolitan cities compared to those living in villages, towns, and cities, participants who perceived their income less than their expenses compared to those who perceived it as equal or higher had higher SCBS scores (Table 2).

Pearson's correlation analysis analyzed the correlation between the EAS and SCBS mean scores. Accordingly, a positive, moderate, and statistically significant correlation was found between the two scales (r = 0.433, *p* = 0.001).

The linear regression model created to identify the independent variables predicting the sustainable consumption behaviour levels of nursing students found no autocorrelation between the variables. The created model was found to be significant, linear, and good (F = 23.967, *p* = 0.001, Durbin Watson = 1.946). It was revealed that 20.4 % of the change in sustainable consumption behaviours levels of nursing students was only related to changes in the environmental awareness scale score (*p* = 0.01). The standard deviation change in the sustainable consumption behaviour level was due to the 0.572 change in the environmental awareness level (Table 4).

# DISCUSSION

The results of this research provide evidence for the environmental awareness and sustainable consumption behaviours of nursing students and make a valuable contribution to the literature by developing effective strategies for sustainable production and consumption which is its most significant aspect. These constitute the strength of the study.

In our study, nursing students had high levels of environmental awareness and a moderate-good levels of sustainable consumption behaviour. A positive, moderate, and statistically significant correlation was found between environmental awareness and sustainable consumption behaviour levels of the nursing students.

As in our study, other studies e studies have indicated that young people studying in the field of health have higher environmental awareness compared to young people studying in other fields (social, art, engineering, *etc.*) (*Yapici et al., 2017*; *Arshad et al., 2021*; *Moody-*

*Marshall, 2023*). The high environmental awareness of nursing students is associated with the fact that the nursing education curriculum is based on the concepts of human, health, disease and environment. These results reveal the importance of including environmental issues in the education years.

Similar to our study, *Ahamad & Ariffin (2018)* detected that although the majority of university students had knowledge of sustainable consumption, they exhibited moderate sustainable consumption behaviours. Although the level of knowledge about sustainability is high, the rate of conversion into behaviour is lower. *Vicente-Molina, Fernández-Sainz & Izagirre-Olaizola (2018)* determined that green/environmental product purchasing behaviours were exhibited at the lowest rate. Green and environmentally friendly products are more expensive and therefore younger generations cannot buy these products even though they are more motivated by environmental and social reasons (*Casalegno, Candelo & Santoro, 2022*). Considering Maslow's hierarchy of basic needs, while the priority of individuals with low income is needs such as food, shelter and clothing, achieving the desire for a better world may not be the primary goal (*Li et al., 2021*). Nursing students in Türkiye are generally children of families with a low socio-economic status (*Başkale & Serçekuş, 2015*). For this reason, the sustainable consumption behaviour level of nursing students may be at a moderate level.

In our study, the regression model for environmental awareness determined that nursing students' gender, the people the students lived with, and the SCBS score were statistically predictive variables. Both our study and other studies in the literature identified that females had higher environmental awareness, environmental protection, and product use awareness than males (*Yapici et al., 2017*; *Vicente-Molina, Fernández-Sainz & Izagirre-Olaizola, 2018*). Active efforts of the ecofeminist movement to prioritize gender in environmental and climate change issues since the 1990s can explain the fact that women are more sensitive to the environment in the results obtained from both our study and other studies in the literature (*Liobikienė, Mandravickaitė & Bernatonienė, 2016*; *Samwel & Muradashvili, 2021*). The fact that the members of the nursing profession mostly consist of females may also be the reason for the high environmental awareness of students in our study. This should be regarded as an opportunity.

*Della Valle (2019)* noted that place of abode and self-efficacy are influential factors of environmental practice. In our study, the people nursing students lived with caused a significant difference in their environmental awareness. It was determined that students living alone had lower environmental awareness compared to those living with their families. The reason for this may be that people living together provide social control by warning others.

The regression model created to determine the predictive variables for sustainable consumption behaviour determined that the only predictive variable that caused a statistically significant difference was the EAS score. The lack of a predictive effect of other sociodemographic variables of nursing students reveals the importance of increasing sustainability and environmental awareness through education. Environment and sustainability issues which are universal, should be included in nursing curricula in a standard way in terms of credit and content both in Türkiye and other countries.

## CONCLUSIONS

Nursing students in this study had high environmental awareness and moderate to good sustainable consumption behaviours. There was a relationship between environmental awareness and sustainable consumption behaviours of nursing students. The most predictive variables in the environmental awareness of nursing students were their gender, the people they lived with, and their sustainable consumption behaviours. It was revealed that environmental awareness level was a significant predictor variable in sustainable consumption behaviours. The effect of especially demographic variables on the sustainable consumption behaviour of nurse students was limited.

An integrated form of a curriculum can be developed to raise responsible environmental practice skills in young people especially at the higher education level, as well as nursing students for the desired environmental and sustainability awareness. For the desired behaviours to occur, it is recommended to examine in other studies why the behaviour and awareness of individuals, whose effects have been revealed in this study, differ according to gender and the people they live with.

This study has several limitations. First, the study was conducted at a nursing faculty in Türkiye. Despite this, a high rate of participation was achieved, representing every class of the faculty. However, the results of this study can only be generalized to the research group.

Second, the EAS and the SCBS used in this study are generally intended to be used in all disciplines. In the literature, there are no scales for the effects of healthcare services or the nursing discipline. It is necessary to develop scales specific to nursing or healthcare services.

## ACKNOWLEDGEMENTS

The authors would like to thank all participants who so willingly participated in this study.

### Funding

The authors received no funding for this work.

### Competing Interests

The authors declare there are no competing interests.

### Author Contributions

- Renginar Öztürk Dönmez conceived and designed the experiments, performed the experiments, analyzed the data, prepared figures and/or tables, authored or reviewed drafts of the article, and approved the final draft.
- Eda Yardımcı conceived and designed the experiments, performed the experiments, prepared figures and/or tables, and approved the final draft.

### Human Ethics

The following information was supplied relating to ethical approvals (*i.e.*, approving body and any reference numbers):

The study was conducted after obtaining ethical approval from the Ethics Committee of Ege University.

## Data Availability

The data is available in the Supplemental File.

## Supplemental Information

Supplemental information for this article can be found online at http://dx.doi.org/10.7717/peerj.17366#supplemental-information.

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
