# Peer review of "Environmental awareness and sustainable consumption behaviours of Turkish nursing students"

_PeerJ, doi:10.7717/peerj.17366_

## Round 0.1 · original submission · Major Revisions

The authors should cite better journals, replace websites and poor journals with better ones.

No need to separate limitation section, merge that with the conclusion section.
Add conclusion section.

No need to separate the Ethical considerations from the research method, state that either in the research method section or at the end after conclusion.
What do you mean by sustainable consumption behaviours among nursing students?

Why do we need to know this special group? Why Turkey? Can the results be generalised?

The literature review is too short, it fails to identify related papers, say, what have been done in environmental awareness? We have already got some papers, e.g. "The Impact of Sustainability Awareness and Moral Values on Environmental Laws, Sustainability 2021, 13, 5882" How do your papers compare the existing ones is important in identifying the research gap.

Introduce the few sections at the end of the introduction.

a random route sampling method? What is that?

Where do you recruit these students? Do they come from the same university? Which universities have been included? Are they postgraduate or undergraduate students?

Data analysis is too short and unclear, it needs some maths and state where have these been applied" The data were evaluated in the SPSS 25 package program. Number, percentage, mean, Kolmogorov-Smirnov Z test, t-test, one-way analysis of variance, independent t-test, Pearson's correlation test, and linear regression analysis were used in statistical evaluation. The statistical significance level was considered as p<0.05 at a confidence interval of 95%."

Reviewer 1 ·

Basic reporting

Thank you for the invitation to review this article. When I read the introduction and method sections of the study, I concluded that there were confusing parts in the paper, and introduction is not clear and do not contain enough information. Additionally, the manuscript edited by a fluent English speaker.

Experimental design

In the abstract please give information about the profile of participations.
In the introduction, more information is needed regarding environmental awareness and sustainable consumption which are the basic parameters of the study. The information about the previous literature was given undetaily and not showed the importance of this topic. May be the statistics from previous international and national studies could be added. Please use the references at least last five year.

The "research question" included in the purpose of the study should be removed.
Hypotheses should be added instead of reserach question.

*Please explain sample selection?
*How did you calculate the sample size? It is not clear just reference to Saunders et al. (2010).

The study was not reported to CONSORT. Losses in the study should be disclosed.
Ethical criteria is clear and it is well explained.

Validity of the findings

Merit: avarage, adaquate
Impact:infrequently cited

Results:
-Must be rewritten, plase not repeating the table.

Additional comments

I suggest that the authors rewrite the Discussion section. I'd like the results and discussion to be a bit stronger. So, what makes your information new and different? What does this information do to improve nursing students? Most of the discussion paragraphs simply report the findings and then narrate different findings from related studies. What is missing in the discussion section is what these findings mean.

*Please think about limitations. You could add the strengths of this study.

I think the authors must make a more forceful statement in their conclusion.

Reviewer 2 ·

Basic reporting

Dear author/s,
The subject of the research is important. Nurses have an important place among health professionals. It is thought that this research creates awareness about environmental awareness and sustainable consumption in nursing students who will provide health services in the future. Research is important in this respect. It is thought that the authors described the findings of the study well. It is thought that this research will contribute to the literature.

Minor English language correction required.The English language should be improved to ensure that an international audience can clearly understand your text. Some examples where the language could be improved include lines 24, 25, 26, 256, 257– the current phrasing makes comprehension difficult. I suggest you have a colleague who is proficient in English and familiar with the subject matter review your manuscript, or contact a Professional editing service.

Experimental design

In the Summary section of the research, the method used in the analysis of the data should be mentioned.

Validity of the findings

The subject of the research is important. Nurses have an important place among health professionals. It is thought that this research creates awareness about environmental awareness and sustainable consumption in nursing students who will provide health services in the future. Research is important in this respect. It is thought that the authors described the findings of the study well. It is thought that this research will contribute to the literature.

Best regards,

Annotated reviews are not available for download in order to protect the identity of reviewers who chose to remain anonymous.

---

## Round 0.2 · Major Revisions

Thank you for the revision of the manuscript and your patience in the review process. I have been asked to take over as the prior Academic Editor is unavailable.

Although the quality of the manuscript is improved, there are still a number of issues to be solved. As highlighted by reviewer 1, the introduction and discussion should be better structured to improve the flow of the manuscript. In addition, the discussion should focus more strongly on the evaluation of your own results, their limitations and the comparison with the findings of other studies. Paraphrasing the results is neither necessary nor desirable.

Another point is that the information in your Material and Methods section is not specific enough to understand how you proceeded. In addition to a clearer explanation of the sample size formula in line 111, the questionnaires for collecting personal data and the EAS and SCBS questionnaires should be submitted in English translation as supplemental files.

There are also issues with the references (some cited in the text are missing in the respective chapter), their formatting as well as linguistic and logical mistakes. Please refer to the annotated PDF where I have highlighted these and proposed also possible changes.

I look forward to receiving your revised manuscript.

Reviewer 1 ·

Basic reporting

- Many thanks for submitting this manuscript, which covers an important element of the nursing curriculum. The method is presented relatively clearly, table are helpful. However, there are some elements of the work that require revision;
-The work needs to make useful reference to the wider evidence-base related to sustainable development goals and nursing practice.
- My overall opinion is that this article presents interesting and relevant elements concerning environmental awareness and sustainable consumption among nursing students. However, I found some pieces of information written in the paragraphs are confusing, especially in the introduction and discussion sections. This circumstance disrupts the flow of the manuscript and makes critical messages that want to be delivered through this article unclear.
The explanations included in the discussion section are too broad, and some are less relevant. Hence, it is hard to grasp the key messages. Besides, there is more than one main idea in almost every paragraph yet seems disconnected, making it difficult to follow.
- The revisions were not well-done.
- Comment: In the limitation part,. “ Moreover, in Türkiye, only the faculty where the study was conducted offers "environmental health nursing" and "sustainable health services and nursing" courses. For this reason, the results of the research can be generalized.” BUT You can not generalised your results.
- Comment: What do you mean by sustainable consumption behaviours among nursing students? Although the authors made revisions, it is still unclear what the topic means for nursing and why this group is addressed.

Experimental design

- Comment: Please explain sample selection. *How did you calculate the sample size? It is not clear just reference to Erdoğan et al., 2014.

Validity of the findings

- Comment: Why do we need to know this special group? Why Türkiye? Can the results be generalised? This part is still unclear. Why they reserach this topic in Turkiye.

Reviewer 2 ·

Basic reporting

It appears that the author(s) made the corrections to the article that I had suggested in my previous review.

Experimental design

It appears that the author(s) made the corrections to the article that I had suggested in my previous review.

Validity of the findings

It appears that the author(s) made the corrections to the article that I had suggested in my previous review.

Best regards,

Additional comments

The topic of your article is extremely important. I think it will make both practical and theoretical contributions.

Congratulations to the authors.

---

## Round 0.3 · accepted · Accept

Thank you for the revision of the manuscript. I hereby certify that you have adequately taken into account the comments and improved the manuscript accordingly. Based on my assessment as an Academic Editor, your manuscript is now ready for publication.

Reviewer 2 ·

Basic reporting

I previously expressed my opinion when I reviewed the article. The article can be published.
Title: "Environmental awareness and sustainable consumption behaviours of Turkish nursing students"

Best regards,

Experimental design

I previously expressed my opinion when I reviewed the article. The article can be published.
Title: "Environmental awareness and sustainable consumption behaviours of Turkish nursing students"

Validity of the findings

I previously expressed my opinion when I reviewed the article. The article can be published.
Title: "Environmental awareness and sustainable consumption behaviours of Turkish nursing students"

Additional comments

I previously expressed my opinion when I reviewed the article. The article can be published.
Title: "Environmental awareness and sustainable consumption behaviours of Turkish nursing students"